# Evaluating the Environmental Impact of Construction within the Industrialized Building Process: A Monetization and Building Information Modelling Approach

**DOI:** 10.3390/ijerph17228396

**Published:** 2020-11-13

**Authors:** Fuyi Yao, Guiwen Liu, Yingbo Ji, Wenjing Tong, Xiaoyun Du, Kaijian Li, Asheem Shrestha, Igor Martek

**Affiliations:** 1School of Management Science and Real Estate, Chongqing University, Chongqing 400044, China; fy.yao@cqu.edu.cn (F.Y.); xiaoyundu@cqu.edu.cn (X.D.); likaijian@cqu.edu.cn (K.L.); 2Research Center of Construction Industrialization and Integrated Technology (CIIT), Beijing 100144, China; ji_yb@ncut.edu.cn (Y.J.); wj.tong@mail.ncut.edu.cn (W.T.); 3School of Civil Engineering, North China University of Technology, Beijing 100144, China; 4School of Architecture & Built Environment, Deakin University, Geelong Waterfront Campus, Locked Bag 20001, Geelong, VIC 3220, Australia; asheemsh@hotmail.com (A.S.); igor.martek@deakin.edu.au (I.M.)

**Keywords:** evaluation, environmental impact, industrialized building, building information modeling (BIM)

## Abstract

Industrialization has been widely regarded as a sustainable construction method in terms of its environmental friendliness. However, existing studies mainly consider the single impact of greenhouse gas emissions or material consumption in the construction process of industrialized buildings, and pay less attention to ecological pollution and community interest, which leads to an insufficient understanding. There is an urgent need to systematically carry out accurate assessment of comprehensive construction environmental impact within industrialized building processes. Various methods, including face-to-face interviews, field research and building information modeling (BIM), were used for data collection. Four categories selected for the study included resource consumption, material loss, ecological pollution, and community interest. A life cycle assessment (LCA) model, namely input-process-output model (IPO), is proposed to analyze the construction environmental impact of the standard layer of industrialized buildings from four life cycle stages, namely, transportation, stacking, assembly and cast-in-place. The monetization approach of willingness to pay (WTP) was applied to make a quantitative comparison. Results reveal that the assembly stage has the largest impact on the environment at 66.13% among the four life cycle stages, followed by transportation at 16.39%, stacking at 10.29%, and cast-in-place at 7.19%. The key factors include power consumption, noise pollution, material loss, fuel consumption and component loss, which altogether account for more than 85% of the total impact. Relevant stakeholders can conduct their project using the same approach to determine the construction environmental performance and hence introduce appropriate measures to mitigate the environmental burden.

## 1. Introduction

China is experiencing rapid urbanization. The Chinese government plans to increase China’s urbanization level to 60% by 2030 [1]. Consequently, China now supports one of the largest construction industries in the world [2]. However, construction is not an environmentally friendly process. Buildings are responsible for over a third of global energy-related carbon dioxide (CO_2_) emissions, accounting for roughly 33% of the total global CO_2_ emission [3]. Dong and Ng [4] calculated that the carbon emissions amount to 637 kg CO_2_ equivalent per square meter of the gross floor area. In fact, the Chinese construction industry accounted for 20% of the total energy consumption in 2015 [5,6], with the construction stage alone contributing over 70% of the total energy-related CO_2_ emission in the building sector [7]. This has caused serious environmental issues [8,9]. To meet these challenges, the Chinese government has implemented key strategies in energy conservation and pollution reduction, with the industrialization of building construction being a main component of that strategy.

Industrialization is an innovative process in which building components are produced in a controlled environment, transported to the construction site and assembled into buildings [10]. The construction method of industrialized buildings is distinct from conventional construction, in which raw or preliminary processed materials, such as iron, timber and concrete, are transported to the construction site directly and cast in situ. A variety of inter-changeable terms are used to refer to a building that uses industrialized construction technology, such as prefabrication, preassembly, modularization, off-site fabrication, off-site production in other countries, and industrialized building and off-site construction in mainland China. Nowadays, the key question remains as to whether industrialization can really achieve the objectives of controlling the negative environmental impacts of creating a sustainable developmental alternative to the traditional building and urbanization process. There has been interest from academia in attempting to assess the environmental issues related to the construction sector. The life cycle assessment (LCA) method as a universal technology of environmental impact assessment has been widely recognized and applied by scholars, which can be process-based, input–output (I-O) and hybrid. For instance, Dong et al. [11] established an LCA model that determined that one cubic meter of prefabricated concrete reduced carbon emissions by 10%. Hong et al. [12] investigated the life-cycle energy use of prefabricated components and found that the recycling process could achieve 16–24% energy reductions, and save 4–14% of total life-cycle energy consumption. Teng et al. [13] found an embodied energy reduction of 15.6%, and operational carbon reductions of 3.2% were achieved from prefabrication. These studies have confirmed that industrialized construction can provide advantages in energy saving and carbon reduction as compared to site-based traditional method.

Although the environmental impact of industrialized buildings has been studied to some degree from both theoretical and practical perspectives, a comprehensive knowledge of the environmental impact of construction activities remains insufficient. As we all know, industrialization has greatly changed the traditional construction process, with materials manufactured in multiple off-site, environments [10], then requiring logistically complex, long-distance transportation to multiple target sites [12], ultimately impacting a greater extent of the surrounding community fabric than traditional construction does. While several studies evaluating the environmental impact of traditional construction have been reported to date [14], only a limited number of works on prefabricated construction are available. Li et al. [15] focused on construction equipment and ancillary materials. Fuertes et al. [16] developed an environmental impact causal model to improve the performance of construction processes in regard to workplace, equipment, material, worker and task factor. In respect to prefabricated construction, Cao et al. [17] conducted a comparative study finding that prefabricated technology can reduce resource consumption by 20.49%, resource depletion by 35.82%, damage to health by 6.61%, and damage to the ecosystem by 3.47%. Based on the literature review, a research gap can be identified in previous studies that are limited to activities on the construction site itself, but do not include the broader community environmental impacts that are warranted for consideration given that an industrialized process is undertaken across a much larger domain. Moreover, current research lacks a systematic and objectively quantitative assessment of the environmental impact of construction activities as they specifically take place in industrialized buildings.

In creating a comprehensive evaluation of the construction environmental impact of the industrialized building, this paper considers the cumulative impact of four factors: resource consumption, material loss, ecological pollution and community impact. Furthermore, this study employs a building information model (BIM) as an auxiliary data collection tool and uses monetization as a quantitative measure.

## 2. Literature Review

Various agencies have developed methodologies for evaluating the environmental performance of construction, such as the U.S. Green Building Council [18], the Building Environment Assessment Method, the Leadership in Energy and Environment Design, the Evaluation Standards for Green Building, and the Green Building Evaluation System of China, among others [19,20]. However, many of these models mainly uses qualitative scoring methods, which are generally subjective and may not provide a comprehensive analysis. Since the parameters used are different, they cannot be compared directly. As an alternative method, LCA is defined as the “compilation and evaluation of the inputs, outputs and potential environmental impacts of a product or process throughout its life cycle” [13,21]. Previous studies have examined building LCA in different life cycles, ranging from raw material extraction to final disposal. The different life cycles include: from cradle to gate, from cradle to site, from cradle to grave and the closed-loop life cycles from cradle to cradle. The method can quantitatively evaluate the environmental impact of a building based on many recognized impact categories. Moreover, LCA can be divided into process-based [22], input–output and hybrid [23,24]. Considering the complexity of the construction environment and the similarity between characteristics of industrialized buildings and outputs from the manufacturing industry, an input-process-output [25] model (a hybrid LCA-based system) was proposed for this study. The model can analyze and describe the construction environmental impact in the industrialized building process, across the life cycle from transportation through to the construction site.

Moreover, a monetization approach can be adopted to visualize the impact value in the industrialized building process. This is based on the social willingness to pay (WTP) theory in environmental economics [26,27]. In other words, a developer would need to pay a certain amount of money in order to offset the negative impact on the environment, such as energy consumption, ecological damage and pollution, and compensation to an affected community caused by, for example, the impact of additional noise and crowded road conditions. The monetization of carbon emissions can be obtained by calculating the weight of environmental impact factors, calculated as the impact potential factor multiplied by the monetary factor. These data are available in government reports, research papers, and certain databases. For material consumption and community influence, monetization can be calculated using bill of quantities (BOQ) analysis, and through field research.

Additionally, a BIM approach provides an effective platform for overcoming the difficulties of acquiring building data [8]. BIM is established at the design stage, with relevant information added to the model, including material category, reinforcement type, dimensions, volume, weight, supplier, and so on. During the construction process, the BIM model can provide detailed project information to improve management decisions, speed and accuracy.

## 3. Method

To achieve the research objective of systematically carrying out accurate assessment of the environmental impact of comprehensive construction within industrialized building processes, two major research methods are employed in this study: (1) inventory analysis approach; and (2) case study. The inventory analysis application of LCA framework includes four distinct analytical steps: defining the goal and scope, creating the life-cycle inventory, assessing the impact and interpreting the results. After analyzing the characteristics of industrialized building construction, a hybrid LCA-based system is developed with a proposed input-process-output model (IPO) to assessment the construction environmental impacts, and three objectives are addressed: (1) to determine the scope of activities that characterize industrialized buildings, (2) to identify the environmental impact factors; and (3) to quantify those impact factors.

Four steps are carried out to achieve these objectives:(i)Identification of the research boundary;(ii)Proposing the input-process-output model (IPO) of industrialized building construction;(iii)Calculating the evaluation impacts by using a monetization approach;(iv)Exploration of study strategies.

Figure 1 shows the framework of research adopted in this study.

### 3.1. Identification of Research Boundary

A literature review and field survey were used to identify the research boundary of the construction environmental impacts in construction processes for industrialized buildings.

Generally, the construction stage was divided into three sub-processes, including material manufacturing, transportation, and on-site construction work [24]. However, in this research, the construction stage is divided into transportation of components, stacking, assembling and the cast-in-place four stages. The reasons are as follows. (i) The production of prefabricated components is in an off-site factory-based activity [28,29], where the working conditions are controlled for its environmental impacts, and are relatively independent of on-site activities. As a result, the production stage within the off-site factory is not assessed to be within the range of the present research. (ii) The main difference between traditional construction and industrialized buildings relate to the manner of field assembly. Thus, in order to ensure timely execution of the construction schedule, stacking of components becomes essential [4]. Finally, (iii) a cast-in-place stage must be included to ensure structural integrity. Raw materials are the composition of the industrialized building structure itself, which is not considered in the environmental impact evaluation of the production process.

### 3.2. The IPO Frame Model of Industrialized Building

#### 3.2.1. Analyzing the Characteristics of Industrialized Building

Compared with traditional methods of site-based construction, industrialized building has brought out many changes, including the construction process and output performance [30]. Analyzing the characteristics of industrialized buildings is a key precondition step to identify impact factors, which can be summarized as follows: (i) transportation from an off-site factory to the construction site [12]. Prefabricated logistics requiring heavy transport to be used in shipping prefabricated components of large volume and weight, where route planning is required; (ii) vertical and horizontal hoisting using large construction machines is needed to position and fix prefabricated components; (iii) if the construction site is in a community within a city, there will be some disruptive influences such as noise pollution [31], traffic congestion, etc. Overall however, industrialized technology offers a potential reduction in time, environmental impact, and a consequent increase in predictability [32,33].

#### 3.2.2. Proposing the IPO Model

Based on the research boundary analysis of industrialized building, an LCA model was proposed; specifically, the construction processing IPO model of industrialized building, shown in Figure 2. The procedure can be summarized as follows:(1)Analysis of the flow path

The IPO model is divided into three categories defined by the research boundary. These are the input of resources and energy, prefabricated components, and raw materials; the process of assembly construction; the output of a standard layer of an industrialized building. Clearly, this flow path of construction activities will have environmental implications.(2)Analysis of the dynamic exchange system

The construction process involves a dynamic exchange system, requiring significant amounts of manpower and material input, yet the final delivered product does not contain the totality of energy inputs. According to the law of conservation of energy, some energy will be dissipated into the external environment. The external environmental impacts are usually negative, involving such consequences as consumption of resources and energy, losses of materials, ecological pollution, and disruptions to the surrounding cultural or community environment. Ecological pollution is defined as when the ecological environment has been greatly damaged due to pollution; the pollution usually includes global warming, acidification, airborne suspended particles and solid waste [17].(3)Building up the construction processing IPO model

Based on the above analysis, two significant areas can be distinguished: closed and open. The closed part represents the construction process of a standard layer of an industrialized building while the open part represents the environment.

#### 3.2.3. Inventory Analysis of Impact Factors

This research set a four-stage research boundary to analyze the impacts of a standard assembly layer on the external environment during the construction sub-processes. The impact categories were retrieved from the literature review, field research, and face-to-face structured interviews.

The construction environmental impacts were divided into four categories including consumption of resources and energy [18], loss of materials (*L_m_*) [34], ecological pollution (*P_e_*), and adverse effects on the cultural or community environment (*Q_c_*). The inventory analysis of impact can be summarized as follows: (i) in the transportation stage, which refers to the fuel consumption of the transporter, component losses resulting from shipping and improper loading measures, vehicle exhaust, the damage of fixed components. (ii) In order to ensure the smooth implementation of the construction progress some components need to be stacked ahead of schedule. At the stacking stage, the impact factors include component loss of stacking, placing piece loss, fuel consumption of re-handling, occupation of land resources. (iii) In the construction process, the assembling work gives rise to major impacts. The resource inputs include large tower cranes, electric power, professional workers, and others. These inputs impact the environment through such events as prefabricated component loss, fuel consumption, exhaust of vertical cranes, solid waste, noise pollution, etc. (iv) The key node cast-in-place is an important practice for ensuring building safety, and the connection points include composite floor casting and beam column connection. The main environmental impacts include wastewater, fuel consumption and the exhaust of the ready-mix concrete truck.

Finally, a face-to-face structured interview method was adopted to correct and refine these impact factors, by inviting several construction site management personnel to add their expertise.

### 3.3. Calculating the Evaluation Impacts

The construction environmental impact refers to the social and environmental costs of the industrialized construction process, not including a consideration of the building structure itself. The magnitude of the impact is calculated using a monetization method. Based on the construction process, a linear mathematical model is selected to calculate the evaluation result, as expressed in Equation (1).
(1)$$V=V_{1}+V_{2}+V_{3}+V_{4}$$
where *V*_1_ is the total monetary value in the transportation stage, *V*_2_ is the total monetary value in the stacking stage, *V*_3_ is the total monetary value in the assembly stage, *V*_4_ is the total monetary value in the cast-in-place operation stage.

#### 3.3.1. Transportation Stage

The prefabricated components are manufactured in an off-site factory. A transportation plan of the construction schedule is necessary, which needs to consider the location of the off-site factory, transport route, fixed support scheme, transport machinery, and the road conditions.

(1) Fuel consumption of the transporter (*A*_1_) is considered to be the most important part in respect to carbon emissions [24]. It is related to the transportation distance, the unit fuel consumption of the transport vehicle [35], and the road conditions. The monetary value of *A*_1_ can be calculated using the following formula:(2)$$C_{A1}=P_{fu}\sum_{i=0}^{n_{1}}\left\{D_{t}\times \left[P_{v}+\frac{W_{ci}}{W_{f}}\left(P_{fv}-P_{v}\right)\right]\times \left(1+k_{r}\right)\right\}$$
where *D_t_* is the transportation distance; *P_v_* is the unit fuel consumption of transport vehicles at no load; *P_fv_* is the unit fuel consumption at full load; *W_ci_* is the weight of the precast components in the single batch *i*; *W_f_* is the full load of the transport vehicle; *k_r_* is the roughness of transportation road; *P_fu_* is the unit price of oil; *n*_1_ is the number of transportation.

(2) Component loss of transportation (*A*_2_) is caused by improper fixed measures or bad road conditions. The losses can be divided into two types: minor repairs, with no need to return to the factory, and heavy repairs, with a necessity to return to the factory. This paper considers the heavy repair situation using the following formula:(3)$$L_{A2}=\sum_{j=0}^{m}\left[\frac{2W_{j}\times C_{A1}}{W_{c}}+\sum_{x=0}^{t}\left(P_{cxj}\times Q_{cxj}\right)\right]$$
where *W_j_* is the weight of damaged components; *P_cxj_* is the unit price of material *x* of damaged component *j*; and *Q_cxj_* is the consumption of material *x*.

(3) Automobile exhaust of the transporter (*A*_3_) includes CO_2_, and NO_x_. As these are the major exhaust components, for simplicity, this paper only considers these two. The energy consumption and emissions inventory analysis used are as determined by the environmental emissions factors listed by the Intergovernmental Panel on Climate Change [36]. Moreover, the carbon trading price used is as per the data provided by China’s carbon emissions trading network of 2018. Thus, the monetary value can be calculated using following formula:(4)$$P_{A3}=P_{c}\times \left(k_{\mathrm{CEF-C}O_{2}}+k_{\mathrm{CEF-N}O_{X}}\right)\times \sum_{i=0}^{n_{1}}\left\{D_{t}\times \left[P_{v}+\frac{W_{ci}}{W_{f}}\left(P_{fv}-P_{v}\right)\right]\times \left(1+k_{r}\right)\right\}$$
where *P_c_* is the carbon trading price, set at CNY 51/ton by end of 2018; *k_CEF_* is the carbon emission monetary factor value, and according to “IPCC Guidelines for National Greenhouse Gas Inventories (1996)”, the value of *k_CEF-CO_2__* is 0.7623, and the value of *k_CEF-NO_x__* is 0.9000.

(4) Fixed components consumption (*A*_4_) refers to the consumption of the fixed steel frame on the component transporter. The monetary value can be calculated using following formula:(5)$$C_{A4}=\frac{P_{s}\times \left(1+r_{s}\right)}{n_{3}}\times \sum_{i=0}^{n_{2}}Q_{i}$$
where *P_s_* is the unit price of steel in the construction area; *r_s_* is the scrap rate of steel; *Q_i_* is the amount of steel in the transporter *i*; *n*_2_ is the number of transporters; *n*_3_ is the number of standard layers.

(5) Traffic jams (*A*_5_) is an indicator of social ecological impact. The monetary value can be interpreted as the willingness to pay to reduce traffic jams over the component’s transporter routes [37]. It can be calculated using following formula:(6)$$Q_{A5}=D_{t}\times \left(Q_{zt}-Q_{z}\right)\times WTP_{t}$$
where *Q_zt_* is the traffic flow on transport lines; *Q_z_* is the traffic flow in the absence of component transportation; *WTP_t_* is the “willing to pay” amount for reducing the traffic jams across every kilometer within a community.

Thus, the total monetary value in the transportation stage (*V*_1_) can be calculated using following formula:(7)$$V_{1}=C_{A1}+L_{A2}+P_{A3}+C_{A4}+Q_{A5}$$

#### 3.3.2. Stacking Stage

In order to ensure the integrity of the construction schedule, it is necessary to deliver a certain number of prefabricated components in advance. However, adding this intermediate link between the transportation and assembling stage will create negative impacts.

(1) Component loss or damage due to stacking (*B*_1_) are accidental contingency events. Based on field studies, the extent of damage can be divided into two types. The first requires heavy repairs, where components need to be returned to the factory. The second involves local small-scale damage, where components can be repaired on the construction site. This paper considers heavy damage, and the monetary value can be calculated using following formula:(8)$$L_{B1}=\frac{1}{f}\sum_{j=0}^{m}\sum_{x=0}^{t}\left(P_{cxj}\times Q_{cxj}\right)$$
where *f* is the conditional probability of heavy repairs compared to site repairs, derived from estimations offered by experienced workers through field study.

(2) Placing piece loss (*B*_2_) refers to the loss of items used in fixing construction components into the building. This paper hypothesizes that a placing piece is disposable and need not be replaced. So, the monetary value can be calculated using following formula:(9)$$L_{B2}=\frac{P_{s}}{n_{3}}\times \left(1-k_{1}\right)\times \sum_{j=0}^{m_{1}}Q_{j}$$
where *k*_1_ is depreciation rate of wasted steel; *m*_1_ refers to the type of placing piece.

(3) Fuel consumption of components re-handling (*B*_3_) refers to the consumption required for short distance small mechanical transport. The machinery types include forklifts and load transport vehicles, with the assumption that vehicles are always operate in a full-load condition. So, the formula can be followed:(10)$$C_{B3}=2P_{fu}\times \sum_{i=0}^{n_{4}}\left(D_{\mathrm{t-ri}}\times P_{\mathrm{fv-ri}}\right)$$
where *D_t-ri_* is the distance in kilometers travelled regarding short distance re-handing; *P_fv-ri_* is the unit fuel consumption at full load in the re-handing process; *n*_4_ is the number of re-handling events.

(4) Occupation of land resources (*B*_4_) refers to the land resource used at the stacking stage. This indicator can be measured by the opportunity cost of land use, which takes on the unit price of local warehousing as the equivalent. So, the monetary value can be calculated using following formula:(11)$$Q_{B4}=S_{s}\times T_{1}\times \frac{k_{2}\times P_{oc}}{30}$$
where *S_s_* is the area of the prefabricated components stacking yard; *T*_1_ is the construction time of a standard layer; *k*_2_ is the conversion coefficient, as given by the given by developer; *P_oc_* is the opportunity cost for renting a local warehouse.

(5) Exhaust of the handing vehicles (*B*_5_) refers to the vehicle exhaust generated in the re-handing process. So, the monetary value can be calculated using the formula:(12)$$P_{B5}=2P_{c}\times \left(k_{\mathrm{CEF-C}O_{2}}+k_{\mathrm{CEF-N}O_{X}}\right)\times \sum_{i=0}^{n_{4}}\left(D_{\mathrm{t-ri}}\times P_{\mathrm{fv-ri}}\right)$$

Thus, the total monetary value in the stacking stage (*V*_2_) can be calculated using following formula:(13)$$V_{2}=L_{B1}+L_{B2}+C_{B3}+Q_{B4}+P_{B5}$$

#### 3.3.3. Assembling Stage

Assembling work is a complex and systematic activity impacting directly on the construction environment in such areas as power consumption, solid waste generation, machine wear, noise pollution, and the safety of workers.

(1) Component loss of hoisting (*C*_1_) is caused by improper or incorrect operation, such as occurs when using an incorrect lifting point, in large hoisting undulation, heavy wind, or as a result of accidents, all of which may have serious outcomes. So, the monetary value can be calculated using following formula:(14)$$L_{C1}=\sum_{j=0}^{m}\sum_{x=0}^{t}\left(P_{cxj}\times Q_{cxj}\right)$$

(2) Power consumption of assembly (*C*_2_) refers to the consumption of large tower cranes for industrial use, and other machines. The electric consumption can be obtained using an electricity meter. So, the monetary value can be calculated using the following formula:(15)$$C_{C2}=P_{e}\times \sum_{i=0}^{n_{5}}\left(P_{ci}\times H_{ei}\right)$$
where *P_e_* is the local price of electricity; *P_ci_* is the engine power of machine *i*; *H_ei_* is the total working time; *n*_5_ is the number of machines.

(3) Solid waste produced as a result of assembly (*C*_3_) refers to the construction waste generated from reinforcement, formwork and concrete works. So, the monetary value of *C*_3_ can be calculated using following formula:(16)$$P_{C3}=S_{h}\times Q_{w}\times P_{w}$$
where *S_h_* is the covered area of a standard layer; *Q_w_* is the quantity of refuse produced per unit area; *P_w_* is the construction waste disposal fee, which in Beijing is priced at CNY 40 per ton [38].

(4) Consumption of mechanics (*C*_4_) refers to machine maintenance and depreciation charges. The data used were obtained from field research, with depreciation charges calculated using a units-of-production method. So, the monetary value can be calculated using following formula:(17)$$C_{C4}=\sum_{i=0}^{n_{5}}\left[F_{mi}\times \frac{T_{1}}{T}+\frac{Q_{si}}{Q_{ti}}\times P_{oi}\times \left(1-r_{oi}\right)\right]$$
where *F_mi_* is maintenance charge of mechanical machinery *i*; *T* is the construction period of the structure; *Q_si_* is the workload in a standard layer of construction for mechanical machinery *i*; *Q_ti_* is the total workload in over the construction period of major structure for the mechanic machinery *i*; *P_oi_* is the lease or hire price of fixed assets of the mechanical machinery *i*; *r_oi_* is the estimated net salvage value rate.

(5) Noise pollution (*C*_5_) refers to machine engine, installation, material processing noise, and the like. Quantification of noise pollution is problematic, and this paper uses the WTP method to calculate the monetary value. It can be calculated using following formula:(18)$$Q_{C5}=M_{n}\times \sum_{i=0}^{n_{6}}\left(F_{ni}\times WTP_{ni}\right)$$
where *M_n_* is days of compensation payment; *F_ni_* is the number of households at *i* distance; *WTP_ni_* is the amount of money a family is willing to pay, at *i* distance; *n*_6_ is the number of different kinds of distances.

(6) The safety of workers (*C*_6_) focuses on the construction risk arising from the need to lift large volumes of prefabricated components. This paper uses the WTP method to measure the safety of (risk to) workers, using the concept of risk exposure [39,40]. So, the monetary value of workers safety can be calculated as following formula:(19)$$Q_{C6}=P_{uo}\times L_{uo}\times \left(1+k_{uo}\right)$$
where *P_uo_* is the probability of occurrence of a safety risk; *L_uo_* is potential loss value; *k_uo_* is the environmental complexity risk coefficient.

Thus, in summary, taking into account the whole set of above considerations, the total monetary value at the assembly stage (*V*_3_) can be calculated using following formula:(20)$$V_{3}=L_{C1}+C_{C2}+P_{C3}+C_{C4}+Q_{C5}+Q_{C6}$$

#### 3.3.4. Cast-In-Place Stage

The main impact of cast-in-place processes in industrialized buildings arises from wastewater, fuel consumption, and vehicle exhaust. The magnitude of the impact is calculated as follows.

(1) Wastewater (*D*_1_) refers to the water resources needed for construction processes, along with water resources used by construction machinery. The consumption can be obtained through water-meter statistics. The monetary value can be calculated as the following formula:(21)$$P_{D1}=\frac{T_{1}}{T}\times Q_{w}\times P_{w}$$
where *Q_w_* is the water consumption of the structure under construction; *P_w_* is the unit price of industrial water.

(2) Fuel consumption of a concrete transporter (*D*_2_) resulting from the use of commercial concrete. The monetary value can be calculated using following formula:(22)$$C_{D2}=2P_{fu}\times P_{\mathrm{fv-t}}\times D_{t}\times \prod \left(\frac{Q_{tc}}{Q_{c}}\right)$$
where *P_fv-t_* is the unit fuel consumption of a concrete transporter; *Q_tc_* is the total concrete consumption in a standard layer; *Q_c_* is the unit transportation volume of concrete.

(3) Vehicle exhaust of a concrete transporter (*D*_3_). The monetary value *D*_3_ can be calculated using following formula:(23)$$P_{D3}=2P_{\mathrm{fv-t}}\times \prod \left(\frac{Q_{tc}}{Q_{c}}\right)\times P_{c}\times D_{t}\times \left(k_{\mathrm{CEF-C}O_{2}}+k_{\mathrm{CEF-N}O_{X}}\right)$$

Thus, the total monetary value of the cast-in-place operation stage (*V*_4_) can be calculated using the following formula:(24)$$V_{4}=P_{D1}+C_{D2}+P_{D3}$$

### 3.4. Exploration of Study Strategies

The final stage involves understanding the impact results that exist in the construction environment of an industrialized building, through examination of a case study. Corresponding strategies for identifying the key environmental factors are proposed and discussed to address real-world problems.

A representative industrialized building requires a high assembly rate, and is located in an established community. The structure is shear wall structure, which is widely used in the promotion of industrialization construction in China. The types of prefabricated components used include prefabricated walls, prefabricated laminated slabs, prefabricated stairs and prefabricated air conditioning panels.

Fulfilling these requirements, a project was selected, and its general contractor is China State Construction Engineering Corporation (CSCEC). The project locates in the Montougou district of Beijing and exhibits the appropriate complex construction environment necessary for case study. The main parameters of each environmental impact factor were collected from both first and second-hand data: (i) first-hand data included the prefabricated rate and covered, and obtained from the project department. (ii) Second-hand data included social willingness-to-pay price and monetization factors, and were retrieved from databases, official and government websites, as well as published papers.

## 4. Data Collection

### 4.1. Case Profile

The project consists of five buildings with a designed lifespan of 50 years. In consideration of the construction schedule and data availability, this study looks at building #2, and specifically at the twelfth level standard layer as the evaluation unit. Detailed information is presented in Table 1.

### 4.2. Data Collection

According to the construction regulations of Beijing, the dust level needs to be highly controlled in a construction site in order to reduce the haze environment, the studied project applied various environmental protection strategies, such as floor wetting, car washing, and use of an automatic sprinkler system. Thus, the impact of dust pollution is not within the scope of consideration.

The data on impact factors were collected in three ways. The first was through face-to-face interviews with the three project managers, using a structured questionnaire. The second was collecting detailed data, such as electricity and water consumption, and noise pollution, through field research. Five postgraduate students were assigned to the site for a three-month stay, with the data of the whole process of a standard layer from start to finish being tracked and recorded. The last was to use a BIM model to export the component information.(1)Data collection from the BIM model

The building BIM model, LOD 500, was constructed including both a prefabricated part and cast-in-place part, achieved by using Chinese Luban software based on the BIM platform (shown in Figure 3). Additionally, some building data can be collected using the automatic statistics and export function, including engineering quantity information, component types, three-dimensional size, as shown in Table 2.(2)Data collection from field research

Adopting face-to-face interviews and a field research method, field data were collected on resources consumption, material loss, and community impact. The field research was conducted over a time span of three months, with five postgraduate students staying on the construction site each day for that period. The resulting collected data include: (i) statistics of the damaged prefabricated components (shown in Table A1), (ii) construction organization task statistics of the twelfth level standard layer (shown in Table A2), and (iii) prices of all kinds of materials or resources used in construction process and others (shown in Table A3).

## 5. Interpretations of Results

By using Formulas (1)–(24), and the data presented in Section 3, the value of resource consumption, ecological pollution, material loss, and the four aspects of community impact, can be calculated for each stage of the life cycle, for a standard layer of construction in the case study project, shown in Table 3. The specific impact factors include CO_2_, NO_X_, fuel consumption, component loss, power and water consumption, noise pollution, worker safety, and traffic jams.

As seen in the transportation stage, resource consumption and material losses constitute the major proportion at 66.32% and 27.81% of the single stage environmental impact, respectively. Community accounts for only 5.45%, while ecological pollution account for a mere 0.42%. Overall, the total cost is CNY 3305.

In the stacking stage, resource consumption, ecological pollution and community questions contribute much less, at 2.32%, 1.13% and 3.86%, respectively. Material losses, however, contribute the major part with 92.69% and account for CNY 1922.

In the assembly stage, resource consumption and community question occupy a large proportion, at 49.03% and 45.82%, respectively, while other types only contribute about 5.15%. Additionally, the environmental cost is the largest compared with other stages, at 66.13%, or CNY 13,330.

Finally, in the cast-in-place operation stage, resource consumption is the major contributor, accounting for 84.71%, with ecological pollution at 13.29%. The other types of construction environmental impact are equal to zero.

## 6. Discussion

Compared with traditional site-based construction, the industrial construction has an obvious advantage of environmental benefits, usually with a 20% reduction in energy consumption, a 35% reduction in resource depletion and a 3.47% reduction in ecosystem damage

Using the monetization approach, the total construction environmental impact is calculated to be CNY 20,157 for a standard layer of an industrialized building process, which is the compensation value caused by the construction activities. The detailed overall result is shown in Figure 4, which indicates that the assembly stage has the largest impact on the environment at 66.13%, which is an intuitive result given that assembly is the major part of a construction process [16]. Next, the transportation stage comes in at 16.39%, followed by stacking at 10.29%, and cast-in-place at 7.19%. Most of components were pre-fabricated in an off-site factory, greatly increasing the transport workload. Cao et al. [17] studied that transportation work of industrialized buildings increased by 57.28% compared with traditional construction. Consequently, transportation is considered as one of the critical stages in evaluating the environmental impact of the industrialized building process compared to stacking and cast-in-place stages.

From the perspective of the construction process, four stages generate different influencing factors along with their roles. The impact distribution across a single stage is shown in Figure 5. It reveals that resource consumption accounts for the largest portion in the transportation, assembly and cast-in-place stage, at around 50%. This is far behind other construction results, indicating that applying new materials, designs or technologies may decrease the environmental impact [15]. The material losses are high in regard to transportation (28%) and stacking (93%). This is because transportation and re-handling of prefabricated components may result in destructive losses, and many ancillary materials are used with fixed and stacked components, such as steel I-beams, that need to be maintained. The community impact is prominent in the construction process of industrialized building, accounting for 46% at the assembly stage, though only 6% at the transportation stage. This arises from the operating noise of large machinery, increased traffic density and security risks, all causing a negative environmental burden. At the cast-in-place stage, however, the community questions and material losses both accounted for 0%, since, of course, this study is looking at industrialized buildings.

Regarding single construction environmental impact factors, Figure 6 shows that resource consumption is the main environmental impact contributor with a monetization of CNY 9561, for a standard layer of construction. This is consistent with previous studies focused on resource depletion and energy consumption [17]. Community impact, however, needs to be given more attention in regard to the industrialized building construction process, since it is usually ignored in traditional building construction. Even so, it is evident that this problem has been magnified in industrial buildings, and more attention must be paid if sustainable development is to be achieved. By contrast, ecological pollution is not a prominent problem, as the major processing activities of components were completed before entering the construction site. Finally, material losses refer to prefabricated component destruction caused by sub-optimal managerial practices. This can be reduced by strengthening management.

As shown in Figure 7, the key factors of environmental impact can be identified, along with their values, among which the power consumption is 28.32%; noise pollution, 25.09%; material loss, 12.46%; fuel consumption, 10.59%; and component loss, 8.74%. Altogether they account for more than 85% of the total impact among the four life cycle construction stages. This high value indicates that effective strategies are worthy of exploration for mitigation; for example, approaches such as setting up sound insulation to reduce the negative impact on neighboring residents. Moreover, worker safety needs attention [15], not only for monetary value, which is set at CNY 1050, but also because of its ethical imperative, ultimately even deciding the success or failure of a project. Finally, the machine loss, wastewater, vehicle exhaust, traffic jams, solid waste and land consumption, also all have an impact on the project, albeit a relatively smaller one. Even so, there remain many opportunities to reduce the construction environmental burden, such as through, for example, regular maintenance of machines, recycling and better utilization of water resources.

In summary, the structure of construction environmental impact in the industrialized building process has changed significantly as compared with traditional building construction processes. Some control strategies to further reduce the impact were pointed out, as were aspects of the industrialized building process in need of further attention.

## 7. Conclusions

Current research on LCA for the construction phrase of industrialized buildings is limited and a consensus on methodology or approaches has not been fully developed. The environmental impact of industrial buildings construction is unclear, and there is a lack of systematic and comprehensive quantitative research. In this paper, a life cycle assessment (LCA) model, namely input-process-output model (IPO), is proposed to analyze the construction environmental impact of the standard layer of industrialized buildings from four life cycle stages, namely, transportation, stacking, assembly and cast-in-place. Four sub-categories of construction environmental impacts including resource consumption, material losses, ecological pollution and community impact were analyzed through a case study. Calculation procedures were presented, by which these impacts for the specific industrialized building project were quantified. The BIM technology was used for case data collection automatically and the monetization approach of willingness to pay (WTP) was applied to make a quantitative comparison. The findings clearly demonstrate that impact can be measured quantitatively by using the proposed IPO model. This serves as an important method to understand the impacts of industrialized construction on the environment. Specifically, it was found that the assembly stage has the largest impact on the environment at 66.13% among the four life cycle stages, followed by transportation stage at 16.39%, stacking at 10.29%, and cast-in-place at 7.19%. Furthermore, it was seen that the key contributors include power consumption, noise pollution, fuel consumption, material and components loss, which altogether account for more than 85% of the total impact. The structure of construction environmental impact in the industrialized building process varies significantly as compared with traditional building construction processes. Relevant stakeholders can conduct their project using this approach to determine the construction environmental performance and hence introduce appropriate measures to mitigate the environmental burden.

Nevertheless, some limitations of this research exist, such as the failure to inform on the scale of the construction system by framing the data with USD or EUR currency. Additional research must be conducted to study: (1) the application of the international databases such as GaBi or Ecoinvent for data collection to clarity the environmental impacts assessment; (2) the construction environmental performance of industrialized concrete buildings with varying proportions of prefabricated components. Furthermore, more typical buildings should be involved in future research to objectively reflect the construction environmental impacts, and the entire lifespan of industrialized buildings can be researched using the proposed IPO model to estimate the comprehensive environmental impacts.

## Figures and Tables

**Figure 1 ijerph-17-08396-f001:**
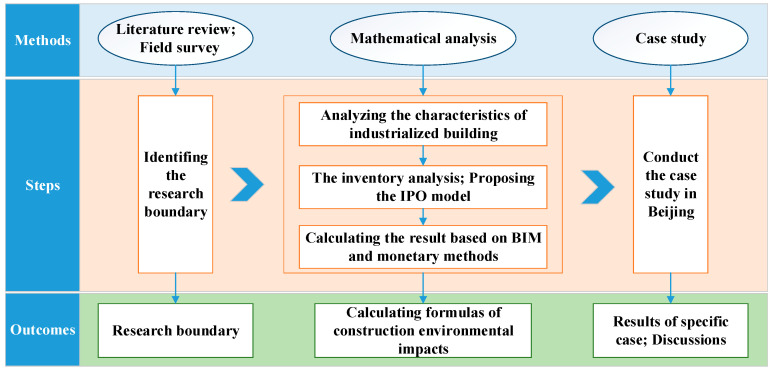
Framework of research method.

**Figure 2 ijerph-17-08396-f002:**
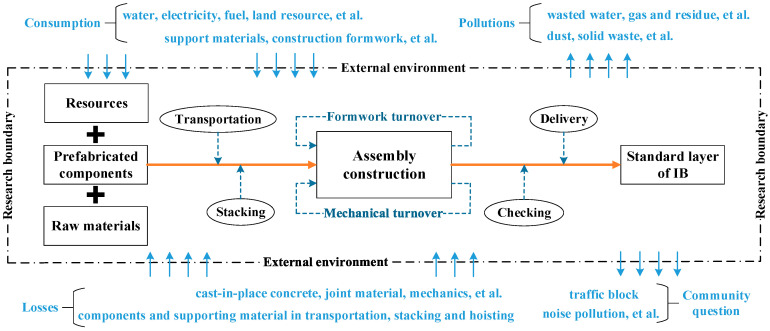
The construction processing input-process-output (IPO) model.

**Figure 3 ijerph-17-08396-f003:**
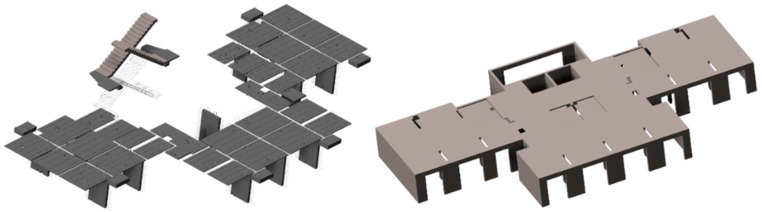
Prefabricated part (left) and cast-in-place part (right).

**Figure 4 ijerph-17-08396-f004:**
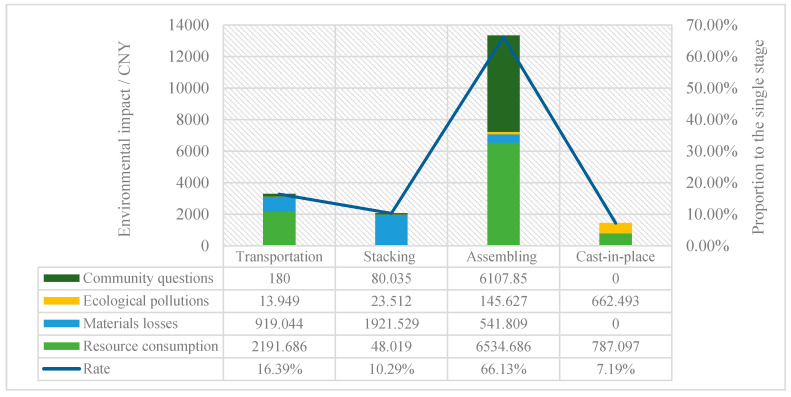
Construction environmental impact of a standard layer.

**Figure 5 ijerph-17-08396-f005:**
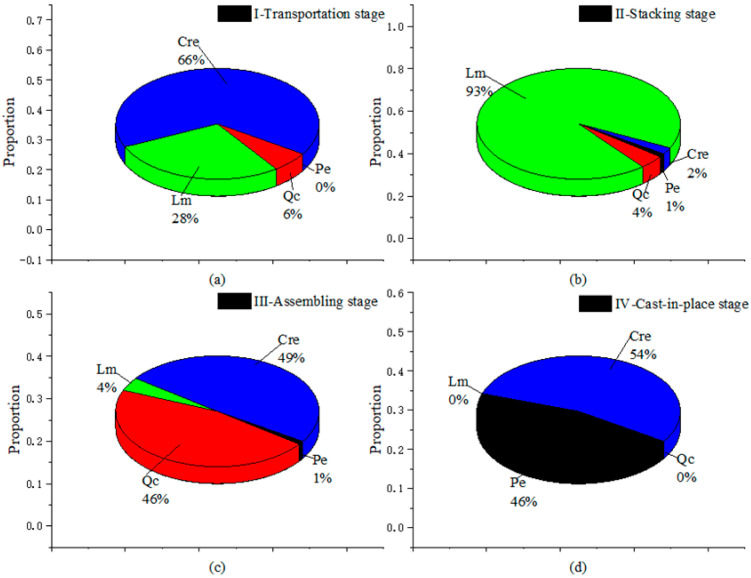
Proportion of different impact categories for each stage. Qc: Community Questions, Cre: Resource Consumption, Lm: Materials Losses, Pe: Ecological Pollutions.

**Figure 6 ijerph-17-08396-f006:**
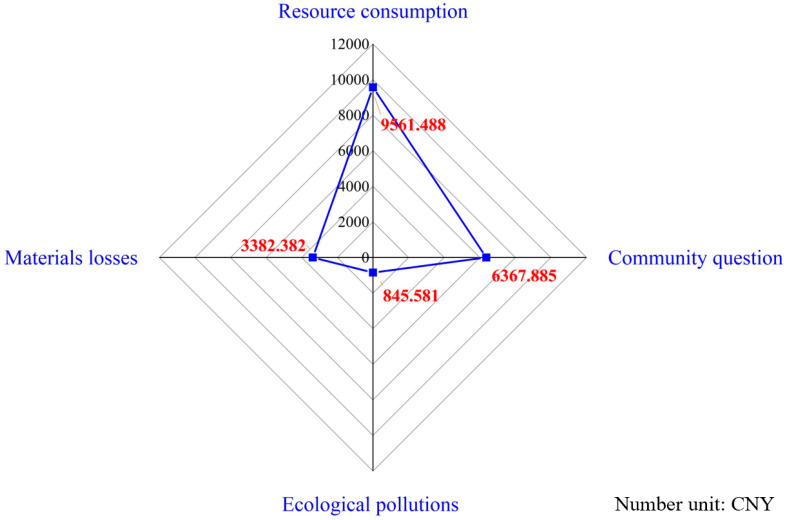
Proportion of construction environmental impact categories.

**Figure 7 ijerph-17-08396-f007:**
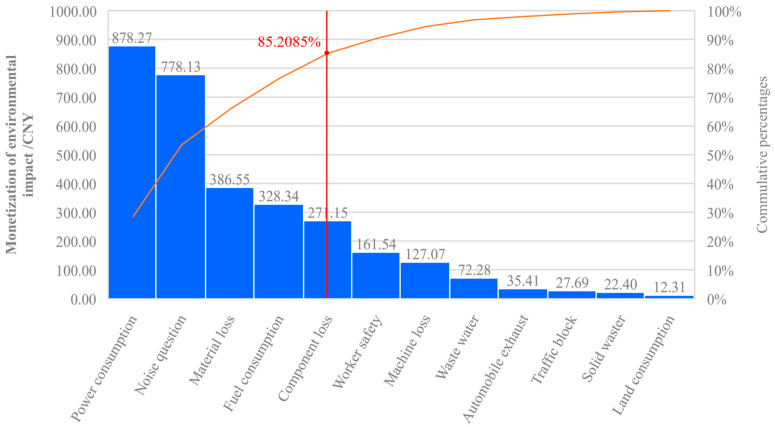
Distribution of construction environmental impact by single substance.

**Table 1 ijerph-17-08396-t001:** Basic information on the industrialized building #2.

Target	Indicators and Descriptions	Units	Parameters
Building type	Resident	-	-
Structure type	Shear wall structure	-	-
Total number of layers	Above ground	layers	25.00
	Underground	layers	4.00
Covered area	A standard layer	m^2^	303.39
Assembly rate	Proportion of precast components	-	0.65
Stacking yard area	For a standard layer	m^2^	200.00
Project duration	Main body structure	d	130.00
	A standard layer	d	6.00
Community population density	The number of people living on land per km^2^ area in the community.	-	975.00
Traffic density	Equal to the ratio of traffic increment to average vehicle flow plus 1	-	1.12
Road condition	Riding Quality Index (RQI)	-	7.50 ^a^
Location of factory	The distance from factory to construction site	km	248.60
Climate influence ^b^	Refers to the project delayed as winter	d	0.00

^a^ When the riding quality is good then 7.0 ≤ RQI ≤ 8.5, and the road properties is Asphalt Pavement. ^b^ The climate of Beijing is temperate monsoon climate and the main structure is assembled in winter.

**Table 2 ijerph-17-08396-t002:** Basic information of main components for the twelfth level standard layer.

Type	NO.	Parameter
Concrete wall	38	V = 27.90 m^3^; L_C_ = C35
Precast concrete air conditioning panel	8	V = 3.86 m^3^; L_C_ = C35
Upward double crooked hook	48	W = 46.08 kg; d = 8 mm
Downward double crooked hook	48	W = 46.08 kg; d = 8 mm
Reinforcement	16	W = 5.60 kg; d = 8 mm
Edge stirrup	34	W = 15.98 kg; d = 9 mm
Edge longitudinal tendons	28	W = 59.64 kg; d = 8 mm
Concrete slab	5	V = 20.53 m^3^; A = 259.43 m^2^; L_C_ = C30
Prefabricated staircase	2	V = 11.60 m^3^; L_C_ = C30
Concrete column	72	V = 30.92 m^3^; L_C_ = C35
Precast concrete walls	12	V = 9.56 m^3^; L_C_ = C30
Precast concrete floor	30	V = 12.24 m^3^; L_C_ = C30
Concrete beam	48	V = 7.42 m^3^; L_C_ = C30

Note: V refers to volume, A refers to area, L_C_ refers to concrete label, W refers to weight, and d refers to diameter. In addition, the steel label is unified as HRB400 or HPB300.

**Table 3 ijerph-17-08396-t003:** Amount and proportion of construction environmental impact in industrialized building process for a standard layer.

Stage	Factor	Resource Consumption (*C_re_*)		Materials Losses (L_m_)		Ecological Pollutions (*P_e_*)		Community Questions (*Q_c_*)	
		Amount (CNY)	%	Amount (CNY)	%	Amount (CNY)	%	Amount (CNY)	%
Transportation (*V*_1_)	Total 1	2191.686	22.92	919.044	27.17	13.949	1.65	180.000	2.83
	*A* _1_	1299.072	39.31	-	-	-	-	-	-
	*A* _2_	-	-	919.044	27.81	-	-	-	-
	*A* _3_	-	-	-	-	13.949	0.42	-	-
	*A* _4_	892.614	27.01	-	-	-	-	-	-
	*A* _5_	-	-	-	-	-	-	180.000	5.45
Stacking (*V*_2_)	Total 2	48.019	0.50	1921.529	56.81	23.512	2.78	80.035	1.26
	*B* _1_	-	-	301.600	14.55	-	-	-	-
	*B* _2_	-	-	1619.929	78.14	-	-	-	-
	*B* _3_	48.019	2.32	-	-	-	-	-	-
	*B* _4_	-	-	-	-	-	-	80.035	3.86
	*B* _5_	-	-	-	-	23.512	1.13	-	-
Assembling (*V*_3_)	Total 3	6534.686	68.34	541.809	16.02	145.627	17.22	6107.850	95.92
	*C* _1_	-	-	541.809	4.06	-	-	-	-
	*C* _2_	5708.736	42.83	-	-	-	-	-	-
	*C* _3_	-	-	-	-	145.627	1.09	-	-
	*C* _4_	825.950	6.20	-	-	-	-	-	-
	*C* _5_	-	-	-	-	-	-	5057.850	37.94
	*C* _6_	-	-	-	-	-	-	1050.000	7.88
Cast-in-place (*V*_4_)	Total 4	787.097	8.24	0	0	662.493	78.35	0	0
	*D* _1_			-	-	469.799	32.41	-	-
	*D* _2_	787.097	52.30	-	-	-	-	-	-
	*D* _3_	-	-	-	-	192.694	13.29	-	-
Total (Adding total 1 to total 4)		9561.488	100	3382.382	100	845.581	100	6367.885	100

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
