# Peer review of "Evaluating the Environmental Impact of Construction within the Industrialized Building Process: A Monetization and Building Information Modelling Approach"

_ijerph, 2020, doi:10.3390/ijerph17228396_

Round 1

Reviewer 1 Report

In this work the authors present a methodology for evaluating the impact of construction within an industrialized building process. 

Overall there are very valuable insights into the evaluation of environmental impact. The article is presented in a logical and clear manner. However, I would like to make the following suggestions

First. To add a subtopic that addresses the potential risks for implementing this methodology. How is this methodology replicable amont China, Asia and other parts of the world. How are the contingency take into consideration in the model? 

I think it will be helpful to inform on the scale of the construction system by framing the data with US dollar or Eur currency. 

Here are some other considerations:

Line

50. The term "Industrialized" is not thoroughly defined. Could you elaborate on the distinction with conventional construction? There are some initials used or synonyms that I think required to be addressed. I think some of the definitions are in line 63 to 69. But the idea looks disjointed.

80-81. Could the author mention a few possible causes why research has lacked a systematic and objective quantification of the environmental impact? This should be addressed again in the conclusions. 

92. I think the research question requires to be stated explicitly if its quoted here. 

349. How is the price of the water determined?

371. Could you elaborate on the nature of the representative building. How big is the project compared with most of the project in China? Is this building a typical construction project around the world?Does this include mixed use?

467. I think this Figure is not completely legible. The data labels are really small. Could you provide also this information in another format as a supplementary material?

Author Response

Response to Reviewer 1 Comments

Point 1: In this work the authors present a methodology for evaluating the impact of construction within an industrialized building process. Overall there are very valuable insights into the evaluation of environmental impact. The article is presented in a logical and clear manner. However, I would like to make the following suggestions.

Response 1: Many thanks for your time and positive comments. All the issues mentioned are carefully addressed below point-by-point.

Point 2: First. To add a subtopic that addresses the potential risks for implementing this methodology. How is this methodology replicable amont China, Asia and other parts of the world. How are the contingency take into consideration in the model? I think it will be helpful to inform on the scale of the construction system by framing the data with US dollar or Eur currency.

Response 2: Thanks for your insightful comments. The industrialized building mainly is represented the adoption of construction industrialisation and the use of prefabrication of components in building construction. A variety of inter-changeable terms are used to refer to building that uses industrialized construction technology, such as “prefabrication, preassembly, modularization and off-site fabrication in the US; “off-site production” in European countries; “industrialized building” in Malaysia; “prefabrication” in Hong Kong and Singapore; and “industrialized building” and “off-site construction” in mainland China. Thus, this methodology replicable amount China, Asia and other parts of the world for the industrialized building with using the prefabricated concrete components producing in an off-site factory. Under the background of housing industrialization has been increasing rapidly in China, the contingency of this model for evaluating the construction environmental impact of the standard layer of industrialized buildings process can be eliminated. The relevant amendments have been revised in the revised manuscript (line 51-56).

In addition, Considering the difficulty of data modification, we keep the proposal of currency nationalization and write it in the research deficiency (line 534-541).

Point 3: Line 50. The term "Industrialized" is not thoroughly defined. Could you elaborate on the distinction with conventional construction? There are some initials used or synonyms that I think required to be addressed. I think some of the definitions are in line 63 to 69. But the idea looks disjointed.

Response 3: Thanks for your valuable suggestions. The construction method of industrialized buildings is distinct from the conventional construction, in which raw or preliminary processed materials, such as iron, timber and concrete, are transported to the construction site directly and cast in situ. The industrialized building mainly is represented the adoption of construction industrialisation and the use of prefabrication of components in building construction. We have added the definitions of the term “Industrialized” in the revised manuscript (line 50-56). The initials used or synonyms also have been addressed (line 26).

[1]  Cao, X. Y., Li, X. D., Zhu, Y., & Liu, G. Q. (2012). Environmental Impact Assessment on Industrialized House and Traditional House Construction. Journal Of Cleaner Production, 155-159.

[2]  Din, M. I., Bahri, N., Dzulkifly, M. A., Norman, M. R., Kamar, K. A. M., & Hamid, Z. A. (2012). The adoption of Industrialised Building System (IBS) construction in Malaysia: The history, policies, experiences and lesson learned. Paper presented at the ISARC. Proceedings of the International Symposium on Automation and Robotics in Construction.

Point 4: Line 80-81. Could the author mention a few possible causes why research has lacked a systematic and objective quantification of the environmental impact? This should be addressed again in the conclusions.

Response 4: Thanks for your constructive suggestion. We have addressed the research gap again in the conclusions to make clear the knowledge contributions in this study in the revised manuscript (line 510-512).

Point 5: Line 92. I think the research question requires to be stated explicitly if its quoted here.

Response 5: Thanks for your valuable comments. We have stated the research question explicitly in line 92 in the revised manuscript (line128-129).

Point 6: Line 349. How is the price of the water determined?

Response 6: Thanks for your careful check and valuable comments. The price of the water mainly selects the price announced by the local government in the case of industrialized building construction, and the detailed information is listed in Table A3.

Point 7: Line 371. Could you elaborate on the nature of the representative building? How big is the project compared with most of the project in China? Is this building a typical construction project around the world? Does this include mixed use?

Response 7: Thanks for your construction comments. The selected representative building is an industrialized housing with a high assembly rate and located in an established community. The building area is 72813 square meters, and prefabricated walls, prefabricated composite boards, prefabricated stairs, prefabricated air conditioning panels are used. The structure is shear wall structure, which is widely used in the promotion of industrialization construction in China. The relevant amendments have been revised in the revised manuscript (line 376-379).

Point 8: Line 467. I think this Figure is not completely legible. The data labels are really small. Could you provide also this information in another format as a supplementary material?

I found that sections 2 & 3 should be re‐organized and be shortened. It may be easier for the readers if the authors define properly the mixture of regression model and the class‐ membership equation first before moving to the computation of the GINI and of the Polarization of subgroups. Sections 2.1 and 2.2 are too long and can be significantly reduced. In section 2.1 the authors assume the condition uk > uj, but this does not appear anywhere else in the calculation of the mixture of regression model. After equation (10) all the other equations are not numbered. 

Response 8: Thanks for your valuable comments. Figure 5 has been resized to make it readable. The section 2 is the core chapter of this paper, which describes the IPO model proposed in this study. The environmental impact of construction within the industrialized building process is stated explicitly to ensures its accuracy and universality. The contents of Section 2 and 3 are adjusted to make the logic and structure of the study more rigorous (line 93-126, and 478).

Reviewer 2 Report

Title:   Evaluating the environmental impact of construction  within the industrialized building process: A  monetization and building information modelling  approach

The paper is very interesting and it is very important for International Journal of Environmental Research and Public Health. However, minor corrections must be taking into account, as follow:

1.- Lines 86-90: Delete these lines as it is repetitive information. With the epigraphs it is already quite clear.

2.-  References: 1, 2, 9, 14, 25, 31, 32 are incomplete according to the regulations of the journal. The initial and final pages would be missing.

3.- Line 613: In reference 40 you capitalize the names of the journal. It should be similar to the other article references such as journal.

Best wishes

Author Response

Response to Reviewer 2 Comments

Point 1: The paper is very interesting and it is very important for International Journal of Environmental Research and Public Health. However, minor corrections must be taking into account, as follow.

Response 1: Many thanks for your time and positive comments. All the issues mentioned are carefully addressed below point-by-point.

Point 2: Lines 86-90: Delete these lines as it is repetitive information. With the epigraphs it is already quite clear.

Response 2: Thanks for your insightful comments. We have deleted the repetitive information in the revised manuscript (line 92-93).

Point 3: References: 1, 2, 9, 14, 25, 31, 32 are incomplete according to the regulations of the journal. The initial and final pages would be missing.

Response 3: Thanks for your careful check. We have completed the missing information in the relevant literature (line 550-554, 568-569, 578-580, 604-605 and 616-619).

Point 4: Line 613: In reference 40 you capitalize the names of the journal. It should be similar to the other article references such as journal.

Response 4: Thanks for your careful check. We have corrected this issue in the revised manuscript (line 634-635).

Reviewer 3 Report

This article evaluates the environmental impacts of construction within the industrialized building process using monetization, BIM, and LCA. The work is in the scope of the journal, however, redaction and structure should be improved as indicated below, especially the methods should be clearer. The author must justify the following points:

Comment 1: This work is missing the academic application of LCA based on ISO 14040 and ISO 14044; Goal and Scope, LCI, LCIA, and Interpretations. The author must differentiate between the midpoint environmental impacts and endpoint environmental impacts. The most important issue herein is to consider other life stages over the entire lifespan of buildings, not only the construction stage. This comes back to the fact that the operation stage is the most complicated stage over the entire lifecycle of buildings, hence the author should consider this stage as long as he is using LCA and BIM.

Comment 2: The scientific contribution is not well-presented in this work. The work has been built based on specific building materials, a limited variant, and a mixing environmental impact. What about other building materials? databases? other lifecycle stages? Moreover, in line 29 it says “Results reveal that the assembly stage has the largest impact on the environment” In Literature is says that the operation stage has the largest impact on the environment over the entire lifespan of buildings. How could the author justify his result?

Comment 3: The author must shed a light on LCA and IPO in the Introduction.

Comment 4: Subsection 2.1. is a Literature Review. It should be relocated in the research and accordingly update the Figure 1.

Comment 5: The author must identify the Functional Equivalent considered for the case study building to evaluate the environmental impacts. This issue is critical and must be justified.  

Comment 6: In section 2.3., the author needs to present the database that IPO uses to model the industrialized building.

Comment 7: In Equation (1), the author has to explain what each variant refers to; V1, V2, V3, and V4. Besides, the presented Equations must be referenced to prove the validity of the study. Additionally, the author must present how the presented Equations, from 1 to 24, have been applied in detail to collect the results.

Comment 8: The presented method of Data Collection in subsection 3.2., is complicated and might be argued against. I would suggest using other international databases for data collection such as GaBi or Ecoinvent. Such an international database could evaluate the clarity of the paper. Particularly, to justify Table 2.

Comment 9: As long as the author is applying LCA methodology, Section 4 should be entitled as "Interpretations". Please find ISO 14040.

Comment 10: The Methods and Discussion Sections are confused and missing lots of necessary details. How resource consumption has been identified? how the community question factor has been evaluated? What is the impact of material loss on the construction market?  Ecological pollution is considered as a midpoint or endpoint category? More questions could appear herein. Hence, the author must well justify these two parts of the study considering the midpoints and endpoints impact categories.

Comment 11: I would suggest rewriting the Conclusions Section based on the following structure where the author needs to highlight the novelty and methods used in this work and point out the collected results. Then, present a summary of the limitations of this research as well as the recommendation for future works. 

Author Response

Response to Reviewer 3 Comments

Point 1: This article evaluates the environmental impacts of construction within the industrialized building process using monetization, BIM, and LCA. The work is in the scope of the journal, however, redaction and structure should be improved as indicated below, especially the methods should be clearer. The author must justify the following points.

Response 1: Many thanks for your time and positive comments. All the issues mentioned are carefully addressed below point-by-point.

Point 2: This work is missing the academic application of LCA based on ISO 14040 and ISO 14044; Goal and Scope, LCI, LCIA, and Interpretations. The author must differentiate between the midpoint environmental impacts and endpoint environmental impacts. The most important issue herein is to consider other life stages over the entire lifespan of buildings, not only the construction stage. This comes back to the fact that the operation stage is the most complicated stage over the entire lifecycle of buildings, hence the author should consider this stage as long as he is using LCA and BIM.

Response 2: Thanks for your insightful comments. LCA is defined as the “compilation and evaluation of the inputs, outputs and potential environmental impacts of a product system throughout its life cycle” [1]. Previous studies have examined building LCA in different life cycles, ranging from raw material extraction to final disposal. The different life cycles include: from cradle to gate, from cradle to site, from cradle to grave and the closed-loop life cycles from cradle to cradle [2]. This study belongs to the midpoint environmental impact assessment and mainly consider the construction processes from four life cycle stages, namely, transportation, stacking, assembly and cast-in-place. In order to make up for the deficiency of this research in the whole life cycle, the lack of research and future prospects have been described in the last part of the conclusion in the revised manuscript. In addition, the application of BIM is mainly the automatic statistical analysis of raw material quantity in inventory analysis in this study.

[1] ISO-14044, (2006). Environmental Management-life Cycle Assessment-requirement and Guidelines, 2006.

[2] Teng, Y., Li, K. J., Pan, W., & Ng, T. (2018). Reducing building life cycle carbon emissions through prefabrication: Evidence from and gaps in empirical studies. Building and Environment, 132, 125-136. doi:10.1016/j.buildenv.2018.01.026.

Point 3: The scientific contribution is not well-presented in this work. The work has been built based on specific building materials, a limited variant, and a mixing environmental impact. What about other building materials? databases? other lifecycle stages? Moreover, in line 29 it says “Results reveal that the assembly stage has the largest impact on the environment” In Literature is says that the operation stage has the largest impact on the environment over the entire lifespan of buildings. How could the author justify his result?

Response 3: Thanks for your valuable suggestions. The selected representative building is an industrialized concrete housing with a high assembly rate and located in an established community. The building area is 72813 square meters, the structure is shear wall structure, and the prefabricated component includes prefabricated walls, prefabricated composite boards, prefabricated stairs and prefabricated air conditioning panels. This is widely used in the promotion of industrialization construction in China. Thus, the selected representative building has a wide range of application. In addition, the main study scope of LCA is to evaluate the construction environmental impact of industrialized building from four life cycle stages, namely, transportation, stacking, assembly and cast-in-place. The calculation results reveal that the assembly stage has the largest impact on the environment among the four life cycle stages. The operation stage is not considered in the environmental impact of construction processes. The expression of the results has been further clarified in the revised manuscript (line 376-379, 384-387).

Point 4: The author must shed a light on LCA and IPO in the Introduction.

Response 4: Thanks for your constructive suggestion. We have shaded a light on LCA in the Introduction. LCA method as a universal technology of environmental impact assessment has been widely recognized and applied by scholars, which can be process-based, input-output (I-O) and hybrid. A hybrid LCA-based system, namely IPO (Input-process-output mode), is proposed to analyse the construction environmental impact of the standard layer of industrialized buildings in this study. Relevant added contents have been added to the revised manuscript (line 60-62).

Point 5: Subsection 2.1. is a Literature Review. It should be relocated in the research and accordingly update the Figure 1.

Response 5: Thanks for your valuable comments. We have relocated the subsection 2.1 as a separate section in this research, namely Section 2. Literature review (line 93-126).

Point 6: The author must identify the Functional Equivalent considered for the case study building to evaluate the environmental impacts. This issue is critical and must be justified.

Response 6: Thanks for your insightful comments. The main innovation and application value of this paper need to be further elaborated. In this study, we selected a standard layer of the representative industrialized building to calculate the construction environmental impact value and quantify its monetary value. Then, the influence of four construction stages on different environmental impact categories is evaluated, and the key factors include power consumption, noise pollution, material loss, fuel consumption and component loss are identified. Therefore, the proposed approach of this study is to guide relevant stakeholders to determine the construction environmental performance and hence introduce appropriate measures to mitigate the environmental burden.

Point 7: In section 2.3., the author needs to present the database that IPO uses to model the industrialized building.

Response 7: Thanks for your constructive comments. In this study, the main parameters of each factor were collected from both first and second-hand data: (i) first-hand data included the prefabricated rate and covered, and obtained from the project department; (ii) second-hand data included social willingness-to-pay price and monetization factors, and were retrieved from databases, official and government websites, as well as published papers. The database of carbon emission monetary factor value is selected as the “IPCC Guidelines for National Greenhouse Gas Inventories (1996)” (line 252-253, 384-387).

Point 8: In Equation (1), the author has to explain what each variant refers to; V1, V2, V3, and V4. Besides, the presented Equations must be referenced to prove the validity of the study. Additionally, the author must present how the presented Equations, from 1 to 24, have been applied in detail to collect the results.

Response 8: Thanks for your valuable comments. We have added the explanation of each variant refers to V1, V2, V3 and V4. Each parameter value in the presented Equations, from 1 to 24, and the basic data of raw materials and energy consumption have been listed in Tables 1, 2, A1, A2 and A3. The results can be obtained by mathematical operation of the presented Equations. Relevant added contents have been added to the revised manuscript (line 221-223).

Point 9: The presented method of Data Collection in subsection 3.2., is complicated and might be argued against. I would suggest using other international databases for data collection such as GaBi or Ecoinvent. Such an international database could evaluate the clarity of the paper. Particularly, to justify Table 2.

Response 9: Thanks for your constructive comments. The presented method of Data Collection in subsection 4.2 mainly includes two aspects: one is to collect the basic building material data by using BIM and field research; the other is to collected the parameter value by a comprehensive data collection method, including existed databases, construction documents and official websites. It is true that this method is complicated, we will explore in the future research to apply some existing databases on the basis of ensuring the accuracy of the data. The future research point has been described in the last part of the conclusion in the revised manuscript (line 534-540).

Point 10: As long as the author is applying LCA methodology, Section 4 should be entitled as "Interpretations". Please find ISO 14040.

Response 10: Thanks for your valuable comments. We have retitled the Section 4 as “Interpretations of results” in the revised manuscript (line 426).

Point 11: The Methods and Discussion Sections are confused and missing lots of necessary details. How resource consumption has been identified? how the community question factor has been evaluated? What is the impact of material loss on the construction market?  Ecological pollution is considered as a midpoint or endpoint category? More questions could appear herein. Hence, the author must well justify these two parts of the study considering the midpoints and endpoints impact categories.

Response 11: Thanks for your constructive comments. First, compared with the traditional cast-in-place operation, the construction process of industrialized buildings is analyzed to understand the construction characteristics. Then, the impact categories are retrieved from literature review, field research, and face-to-face structured interviews. Finally, the negative external environmental impacts are identified from the endpoint category aspect, involving such consequences as consumption of resources and energy, losses of materials, ecological pollution, and disruptions to the surrounding cultural or community environment, and the pollution usually includes global warming, acidification, airborne suspended particles, solid waste, and so on. Finally, each kind of environmental impact categories in each stage is elaborated and calculated in detail in Section 3.3. The relevant contents are marked in the revised manuscript (line 185-189, 196-214, and 215).

Point 12: I would suggest rewriting the Conclusions Section based on the following structure where the author needs to highlight the novelty and methods used in this work and point out the collected results. Then, present a summary of the limitations of this research as well as the recommendation for future works.

Response 12: Thanks for your valuable comments. We have rewriting the Conclusions Section as the suggestions given. The existed limitations of this research and the recommendations for future works also are added in the revised manuscript (line 534-541).

Round 2

Reviewer 1 Report

I think the authors have addressed the observations. However, grammar and spelling should be checked. 

Author Response

Response to Reviewer 1 Comments

Point 1: I think the authors have addressed the observations. However, grammar and spelling should be checked.

Response 1: Many thanks for your time and positive comments. We paid more attention to go through the whole paper with the help of the seventh author (Mr. Asheem) and eighth author (Mr. Igor) who are native speakers in Deakin University. The existed grammar and spelling issues have been corrected in the revised manuscript.

Reviewer 3 Report

The work has developed and the author anserwed most of my previous comments. However, some minor issues are yet to be justified, please. The author needs to clarify the application of LCA framework in Section 3, basically expalin how such framework was applied within IPO model. Moreover, Comment 11 in my previous review needs a better justification, please.

Author Response

Response to Reviewer 3 Comments

Point 1: The work has developed and the author answered most of my previous comments. However, some minor issues are yet to be justified, please.

Response 1: Many thanks for your time and positive comments. All the issues mentioned are carefully addressed below point-by-point.

Point 2: The author needs to clarify the application of LCA framework in Section 3, basically explain how such framework was applied within IPO model.

Response 2: Thanks for your constructive comments. We have addressed the application of LCA framework and explained how such framework was applied within the proposed IPO model. The specific analysis contents about proposed IPO model is in Section 3.2. The revisions also have been highlighted in yellow in the revised manuscript (line 131-144).

Point 3: Moreover, Comment 11 in my previous review needs a better justification, please.

Response 3: Thanks for your valuable comments. Comments 11 in previous reviews is “I would suggest rewriting the Conclusions Section based on the following structure where the author needs to highlight the novelty and methods used in this work and point out the collected results. Then, present a summary of the limitations of this research as well as the recommendation for future works.” We have rewriting the Conclusions as the given suggestions, and highlighting the novelty and methods used in this study and pointed out the collected results (line 516-537). The existed research limitations and the future research works both are identified and added in the revised manuscript (line 538-546).